# OpenReview forum: "SWE-Bench Pro: Can AI Agents Solve Long-Horizon Software Engineering Tasks?"
_ICML.cc/2026/Conference — ICML 2026 regular_

### Official Review · Reviewer_u9Es · 2026-03-08

**Soundness:** 4
**Presentation:** 4
**Significance:** 4
**Originality:** 3
**Overall Recommendation:** 6
**Confidence:** 5

**Summary:**

The paper introduces SWE-Bench Pro, a large scale dataset of SWE problems (1800+) that is a successor to SWE-Bench Verified with improvements like better uniformity across repositories, languages, and higher patch volume. Apart from this the paper adds two privatized subsets: held-out and commercial aimed to combat LLM contaminations with the benchmark ground truth. The benchmark results show significantly higher difficulty of the proposed benchmark compared to SWE-Bench.

**Compliance With Llm Reviewing Policy:**

Affirmed.

**Ethical Review Concerns:**

None.

**Key Questions For Authors:**

[1] Does the private set evaluation harness allow submitting agent code rather than patches? Kaggle Notebook style submission would be valuable for the scientific community to research test-time learning algorithms. The sandbox can and must be devoid of the internet access, the user must provide their own API keys to trusted LLM providers, which should be accessed by the harness directly and the evaluation company must have ZDR and no-training agreements signed.

**Limitations:**

[1] Rolling live updates to the set of tasks are not there, like promised by SWE-Bench Live albeit net delivered.

[2] The commercial dataset has questionable value to the scientific community.

**Strengths And Weaknesses:**

# Strengths

[1] The mission of the paper is important: to have a fresh large scale evaluation dataset for SWE along with the hold-out and private subsets to present a more challenging benchmark than the saturated SWE-Bench Verified that is widely known to be leaked into frontier LLM training.

[2] The results presented in Figure 1 are insightful and encouraging. Better uniformity of the numbers of problems per repo and significantly higher complexity in terms of lines of code is important.

# Weaknesses

[1] I don’t think choosing Copyleft licenses for the basis of the public dataset is strong enough since LLM labs’ crawlers notoriously ignore the licenses. However it is the best that could be done.

---

> ### Author Rebuttal · Authors · 2026-03-31
>
> We thank the reviewer for their thoughtful review and for recommending acceptance. We are pleased that you found the benchmark's mission "important," the results "insightful and encouraging," and that you rated soundness, presentation, and significance all as "excellent." Comments are addressed below
>
> **Copyleft licensing may not be sufficient since LLM labs' crawlers notoriously ignore licenses.**
>
> We agree this is the correct framing, and we do not rely on licensing alone. Our contamination strategy is multi-layered: (1) GPL-licensed repositories provide a legal and best-effort technical barrier, (2) the commercial set uses legally binding exclusive agreements with startups that contractually prohibit redistribution for agent development purposes, and (3) the held-out set is never released publicly. No single layer is sufficient, but together they represent the strongest available contamination controls. We acknowledge that empirical membership inference testing would further validate these claims, and we note this as a direction for future work. We will add explicit language to Section 3.3 making this defense-in-depth argument clearer.
>
> **Rolling live updates are not present, as promised by SWE-Bench Live.**
>
> We acknowledge this limitation. We view a live update mechanism as complementary to our approach: our held-out and commercial sets provide static contamination resistance while a rolling update mechanism would provide temporal freshness. We note this as an important direction for future work and plan to explore a pipeline for periodic benchmark expansion in collaboration with the research community.
>
> **The commercial dataset has questionable value to the scientific community.**
>
> The commercial set serves two distinct purposes: (1) it provides the strongest contamination guarantees of any split, given the exclusive legal agreements, and (2) it allows the community to measure model performance on enterprise-style codebases that reflect industrial complexity and proprietary patterns that are systematically underrepresented in open-source benchmarks. In addition, we believe that a private commercial set serves as a valuable testbed and is a novel evaluation for software engineering agents. While we would like to open-source these repositories, we regrettably cannot due to legal restrictions, but still believe that the inclusion of results and analysis on the set will be a valuable contribution to the scientific community.

---

> > ### Author Rebuttal · Reviewer_u9Es · 2026-04-01
> >
> > Thank you for your answers.

---

### Official Review · Reviewer_UwdG · 2026-03-10

**Soundness:** 3
**Presentation:** 3
**Significance:** 4
**Originality:** 3
**Overall Recommendation:** 5
**Confidence:** 4

**Summary:**

This paper presents SWE-Bench Pro, a new benchmark for long-term, enterprise-level software engineering tasks. The paper makes significant contributions in data collection strategies, task complexity design, and manual validation processes, yet there remains room for improvement in certain aspects.

**Compliance With Llm Reviewing Policy:**

Affirmed.

**Final Justification:**

I appreciate the response and effort in addressing the review concerns. Some of my concerns have been solved.

**Key Questions For Authors:**

* How do stronger agents, such as Trae Agent, perform on SWE‑Bench Pro?
* How does the study ensure that the difficulty distribution of 50–100 instances in each repository is reasonable? Are there repositories whose tasks are systematically more difficult or easier than others?
* The consistency among Public, Commercial, and Held-Out datasets is under question. Are their distributions similar, or do they show noticeable differences?

**Limitations:**

Please see weaknesses.

**Strengths And Weaknesses:**

## Strengths
* **The design of the data contamination protection strategy is reasonable**: The paper adopts a dual anti-contamination mechanism: it constructs public and reserved datasets using repositories under strong copyleft licenses such as GPL, while also incorporating commercial codebases.
* **The task complexity alignins more closely with real-world engineering scenarios**: Benchmarking requires that each task involves modifications to at least 10 lines of code, affecting an average of 4.1 files and 107.4 lines of code. This better reflects the actual challenges encountered in enterprise-level development.
* **The diversity of the benchmark design is commendable**: It covers 41 repositories, multiple programming languages (Python, Go, JavaScript, TypeScript), and various task types (feature addition, bug fixing, refactoring, etc.), with each repository limited to 50–100 instances to prevent overfitting.

## Weaknesses
* **The evaluatied baseline is limited** The paper primarily uses SWE-Agent as the sole scaffold for assessment. Although it mentions that Agentless performs poorly on multi-file editing and is therefore excluded, this decision itself warrants deeper discussion. Different scaffolds may significantly influence model performance, and conclusions drawn based on a single scaffold may not fully reflect the model’s true capabilities.
* **The biases introduced by manual augmentation are not sufficiently discussed**: Although the task descriptions are manually rewritten and enhanced, which improves task quality, they may also introduce systematic bias. The writing style and level of detail provided by different annotators can vary. The paper lacks a quantitative analysis of annotator consistency.
* **The distributional consistency among the Public, Commercial, and Held-Out**: Tables 1 and 2 show that all models perform worse on the commercial set. Interestingly, the ranking order also changes—for example, Claude Opus 4.1 surpasses GPT-5 on the commercial set. How do the authors explain this discrepancy? Does this suggest that commercial codebases present qualitatively different challenges? Furthermore, the authors claim, “We keep this set private to test for overfitting in the future.” How, then, do they ensure that the distribution of the Held-Out set aligns with the other two sets?

---

> ### Author Rebuttal · Authors · 2026-03-31
>
> We thank the reviewer for their thoughtful review and for recognizing the benchmark's strengths in contamination strategy, task complexity, and diversity. We address your weaknesses and questions below.
>
> **Limited scaffold evaluation — SWE-Agent as the sole baseline.**
>
> We have conducted comprehensive evaluations across three agent scaffolds (SWE-Agent, Agentless, and mini-SWE-Agent). The results show consistent performance patterns across all scaffolds, supporting the conclusion that the performance gap reflects model capability rather than scaffold-specific behavior. We will incorporate these results into the final paper.
>
> | Model | SWE-Agent | Agentless | mini-SWE-Agent (bash-only) |
> |---|---|---|---|
> | OpenAI GPT-5 | 23.3 | 22.1 | 21.1 |
> | Claude Opus 4.1 | 22.7 | 21.7 | 20.5 |
> | Claude Sonnet 4 | 17.6 | 18.2 | 17.6 |
> | Gemini 2.5 Pro Preview | 13.5 | 14.3 | 13.6 |
>
> **Biases introduced by manual augmentation — annotator consistency.**
>
> This is a valid concern. To mitigate annotator bias, all augmentations were reviewed by multiple annotators before inclusion, following a structured protocol with explicit guidelines for requirement specificity and interface scope. We will add a paragraph in the dataset construction section describing the annotation protocol, reviewer overlap, and the structured rubric used to standardize augmentation quality across annotators. A quantitative inter-rater agreement analysis will be included as a future improvement.
>
> **Distributional consistency among Public, Commercial, and Held-Out sets — and the ranking reversal between Claude Opus 4.1 and GPT-5 on the commercial set.**
>
> We hypothesize it reflects that the two model families have different relative strengths on enterprise code versus open-source code: GPT-5 may have a stronger prior on open-source repository patterns, while Claude Opus 4.1 may generalize slightly better to novel proprietary codebases. Regarding distributional consistency of the held-out set: we apply the same repository selection criteria, task complexity filters (≥10 LOC), and human augmentation workflow uniformly across all three splits. The held-out set repositories were selected to match the distribution of task types (bug fix, feature addition, refactoring) and programming languages of the public set. We will add a table comparing split-level distributional statistics in the final paper.
>
> **How is difficulty distribution within repositories ensured to be reasonable?**
>
> The 50–100 instance cap per repository is designed precisely to prevent any single repository from dominating the benchmark and to ensure broad coverage. Within each repository, tasks are sampled to cover a range of issue types (bug, feature, refactor) and patch complexity levels. We observe in our per-repository analysis that solve rates do vary across repositories (e.g., qutebrowser achieves ~41% for Claude Sonnet 4, while flitr achieves ~14.5%), which reflects genuine difficulty variation. We do not find a hard correlation with repository size or age, suggesting difficulty is driven primarily by task type and domain specificity rather than codebase metadata alone.
>
> **Are the distributions of Public, Commercial, and Held-Out sets similar?**
>
> The three splits are constructed using the same pipeline and selection criteria, with the primary difference being repository source (open-source GPL vs. commercial vs. held-out GPL). The held-out set is intended to be similar to the public set, with a similar number of repositories and set of problems. The commercial set is on a smaller set of repositories, aiming to be a focused effort on commercial repos.

---

> > ### Author Rebuttal · Reviewer_UwdG · 2026-04-01
> >
> > Thank you for the response. Most of my concerns have been addressed. Possibly due to the word limit, there are some data that were not included in the rebuttal. If these could be provided, I would be happy to raise my score.

---

### Official Review · Reviewer_hk41 · 2026-03-10

**Soundness:** 4
**Presentation:** 4
**Significance:** 3
**Originality:** 3
**Overall Recommendation:** 5
**Confidence:** 4

**Summary:**

This paper introduces SWE-BENCH PRO, a benchmark of 1,865 human-verified and augmented software engineering problems sourced from 41 actively maintained repositories spanning consumer applications, B2B services, and developer tools. The benchmark is partitioned into a public set (731 instances from 11 GPL-licensed repositories), a commercial set (276 instances from 18 proprietary startup codebases), and a held-out set (858 instances from 12 repositories). Key design decisions include: (1) a contamination-resistant sourcing strategy using copyleft-licensed and proprietary code; (2) emphasis on long-horizon, multi-file tasks (average 107.4 LOC across 4.1 files); and (3) a three-stage human-in-the-loop augmentation and verification workflow that adds requirements and interface specifications to mitigate false negatives in unit test verification. Evaluation of widely used coding models under SWE-Agent shows that the best model (GPT-5) achieves only 25.9% Pass@1 on the public set with a further drop to 15.7% on the commercial set. The paper also provides a trajectory failure mode analysis and an ablation study demonstrating the critical role of human augmentations.

**Compliance With Llm Reviewing Policy:**

Affirmed.

**Final Justification:**

good paper

**Key Questions For Authors:**

- The paper presents a valuable comparison by evaluating models on both a public set and a commercial set (sourced from proprietary startup repositories). The reported performance degradation on the commercial set is a compelling finding that successfully highlights the unique challenges of enterprise codebases. However, the analysis of this gap is somewhat brief. What features, or what differences, lead to the gap between private and public datasets? Are commercial tasks harder because of larger codebases, more complex dependency graphs, less community documentation, or domain-specific business logic? Expanding the trajectory failure mode analysis to explicitly contrast the failure distributions between the public and commercial sets would provide deeper insights into the current limitations of LLM agents in industrial settings.
- The ablation study in Section 6.2 on the removal of human augmentations is one of the most interesting parts of the paper. It highlights a fundamental flaw in using existing unit tests for LLM evaluation: real-world software engineering problems often have diverse, equally valid solutions, yet legacy unit tests are rigidly coupled to one specific implementation path. The massive performance drop is striking. However, the paper currently treats human annotation merely as a "fix" for false negatives. To elevate the contribution of this work, the authors should provide a deeper qualitative analysis of the human annotator's role. Specifically: (1) Taxonomy of Human Intervention: What specific types of information were most frequently added by humans to make these "unverifiable" or open-ended problems verifiable? (2) Qualitative Case Studies: While Appendix B provides a good example, an ablation study for LLM trajectory—showing how agent behavior changes with and without specific augmentation types—would help readers better understand the problem.
- The public set relies on legal bounds (copyleft GPL licenses) and privacy restrictions (commercial startup codebases) to claim the benchmark is "contamination-resistant." While this is a logically sound method, it lacks empirical validation. Given the aggressive scraping behaviors of recent LLM pre-training pipelines, license boundaries are often inadvertently crossed.
- The evaluation heavily relies on the SWE-Agent scaffold, with a brief mention that Agentless struggled with multi-file edits. Experiments on more framework may provide more comprehensive results.
- The paper acknowledges in Section 7.1 that "real software engineering tasks may have a variety of correct solutions, even if they do not pass the original tests." This is a fundamental tension that the human augmentation workflow only partially resolves. The augmentations constrain the solution space to reduce false negatives, but in doing so, is there any concerns that penalize creative or unconventional solutions that are functionally correct.
- In March 2026, given the rapid progress in agent capabilities, I am indeed curious about how current SOTA LLMs such as Claude 4.6 Opus perform on the benchmark. However, this will not affect the contribution of this paper.

**Limitations:**

Yes

**Strengths And Weaknesses:**

## Strengths

- Private dataset included
- Extensive human effort for augmentation and verification
- Significantly more difficult tasks
- Addresses saturation of existing benchmarks
- Diverse repository and language coverage
- Well-structured failure mode analysis

## Weaknesses

- Insufficient analysis of public vs. commercial performance gap
- Contamination resistance claim lacks empirical validation

---

> ### Author Rebuttal · Authors · 2026-03-31
>
> We thank the reviewer for their thorough review and recommendation of acceptance, including mentioning the benchmark's strengths including the "extensive human effort for augmentation and verification," the "well-structured failure mode analysis," and the inclusion of a private dataset. We address your questions and weaknesses below.
>
> **Insufficient analysis of the public vs. commercial performance gap.**
>
> Yes, the performance gap between public and commercial sets reflects several qualitative differences. Commercial startup codebases tend to have: (1) denser and more idiosyncratic dependency graphs with fewer publicly documented interfaces, (2) domain-specific business logic that is unlikely to appear in model training data, (3) less community-generated documentation and issue discussion that models might otherwise leverage, and (4) architectural patterns that reflect private internal conventions rather than open-source community standards. We also observe that the ranking between Claude Opus 4.1 and GPT-5 shifts on the commercial set, suggesting that different models have different relative strengths on enterprise versus open-source code. We will expand the failure mode analysis in the final paper to explicitly compare failure distributions between the two sets.
>
> **Deeper qualitative analysis of human augmentation — taxonomy of interventions and case studies.**
>
> We appreciate this suggestion and agree it strengthens the paper's contribution. In terms of a taxonomy, the most frequent human interventions were: (1) adding explicit behavioral requirements that were implied but unstated in the original issue description, (2) specifying interface contracts (function signatures, return types, error handling) to prevent false negatives from valid alternative implementations, and (3) scoping the verifier to exclude tests that checked unrelated functionality. We will add a taxonomy table and expand the qualitative case studies in Appendix B to include trajectory-level ablations showing agent behavior with and without specific augmentation types.
>
> **Contamination resistance lacks empirical validation.**
>
> We agree that license boundaries are not a perfect technical barrier, and we do not claim otherwise. Our contamination strategy is multi-layered: (1) GPL-licensed repositories provide a legal and best-effort technical barrier, (2) the commercial set uses legally binding exclusive agreements with startups that contractually prohibit redistribution for agent development purposes, and (3) the held-out set is never released publicly. No single layer is sufficient, but together they represent the strongest available contamination controls. We acknowledge that empirical contamination testing (e.g., membership inference attacks) would further strengthen this claim, and we note this as a direction for future work.
>
> **Evaluation relies heavily on SWE-Agent.**
>
> We have conducted comprehensive evaluations across three agent scaffolds (SWE-Agent, Agentless, and mini-SWE-Agent), with results showing consistent performance patterns across all three. This addresses the concern that the performance gap might be scaffold-specific.
>
> | Model | SWE-Agent | Agentless | mini-SWE-Agent (bash-only) |
> |---|---|---|---|
> | OpenAI GPT-5 | 23.3 | 22.1 | 21.1 |
> | Claude Opus 4.1 | 22.7 | 21.7 | 20.5 |
> | Claude Sonnet 4 | 17.6 | 18.2 | 17.6 |
> | Gemini 2.5 Pro Preview | 13.5 | 14.3 | 13.6 |
>
> **Human augmentation constraining the solution space and penalizing unconventional correct solutions.**
>
> This is a fundamental tension in benchmark design that we take seriously. Our augmentation strategy is designed to *expand* the verifiable solution space (by reducing false negatives) rather than restrict it. However, we acknowledge that interface specifications may inadvertently constrain the solution to one acceptable API shape in some cases. We discuss this as a limitation in Section 7.1 and view more flexible solution-space verification as an important direction for future work.
>
> **How do current SOTA LLMs such as Claude 4.6 Opus perform?**
>
> We appreciate the curiosity. Given the rapid pace of model development, we agree that running evaluations on new frontier models is valuable and plan to do so as part of ongoing benchmark maintenance. We will report updated leaderboard results as they become available.

---

> > ### Author Rebuttal · Reviewer_hk41 · 2026-04-01
> >
> > Thanks authors for addressing my concerns.

---

### Official Review · Reviewer_cYcP · 2026-03-11

**Soundness:** 3
**Presentation:** 4
**Significance:** 4
**Originality:** 3
**Overall Recommendation:** 5
**Confidence:** 4

**Summary:**

This paper describes the collection of a new coding benchmark suite that aims to address several pressing challenges in AI evaluation. Specifically, the authors collect "problems" from a variety of coding repositories (spanning domains and organization types) and associated solutions such that new models can graded on their long horizon autonomous coding ability and avoid issues around contamination and saturation. The paper motivates the need for new benchmarks (Introduction), briefly covers key related work on benchmarking agents, and then describes the benchmark. Notably, the paper itself provides important info about the pieces of the benchmark that will not be released publicly (for contamination reasons).

**Compliance With Llm Reviewing Policy:**

Affirmed.

**Final Justification:**

The authors provided additional details across the responses to various reviewers that helped to clarify various additional details that can strengthen the work. The rebuttal reinforced my positive view of the paper.

**Key Questions For Authors:**

- How much human labor was involved in validating tasks? Can you provide any descriptive stats on the monetary or time costs for validation, inter-rater reliability, etc.?
- Do you have a sense of compute and API costs required to replicate results?
- What are the institutional concerns that might arise around maintaining private/heldout sets going forward? Are these major concerns for these kinds of work?

**Limitations:**

Brief but reasonable.

**Strengths And Weaknesses:**

In terms of significance, this paper is extremely strong. The draft succeeds in the motivating the value of addressing benchmarking issues and convincing the reader that this new contribution actually does address some of these issues. In general, I don't think the paper has major room for improvement in terms of motivation or arguing for its own significance.

In terms of presentation, the paper itself is structured well and generally communicates key details to the reader well. See below for a few nits and questions that readers may have, mainly around quantitative sense of some of the costs of replicating or doing fresh runs.

In terms of soundness, the main questions here that readers may have are whether the data was sourced in a way that makes it genuinely useful and whether the preparation of "benchmark-ready" data might create any issues (e.g., the task description leak info, if environment setup makes tasks too challenging or too easy). The analysis of results is relatively straightforward and has no major issues. Use of LLM-as-a-judge is reasonable, might be nice to include a bit more detail about human validation and the qualitative analysis here.

Overall, I think this paper describes a significant contribution in a clean and clear manner.

---

> ### Author Rebuttal · Authors · 2026-03-31
>
> We thank the reviewer for the thoughtful review, including mentioning that the paper is "extremely strong" in significance and "excellent" in presentation, and that the benchmark "convincingly addresses" the issues it sets out to solve. Below, we mention the questions asked in the review.
>
> **How much human labor was involved in validating tasks? Can you provide any descriptive stats on the monetary or time costs for validation, inter-rater reliability, etc.?**
>
> SWE-Bench Pro involved over 5,800+ hours of human labor across the augmentation and verification pipeline. Each task went through a three-stage workflow: (1) initial problem augmentation (requirements and interface specification), (2) test environment validation and execution, and (3) a final human verification pass to confirm task resolvability. Inter-rater reliability was enforced via a structured review protocol where a second annotator independently verified each augmented instance before inclusion. We will add a dedicated paragraph in the dataset construction section quantifying these costs and the inter-rater agreement process.
>
> **Do you have a sense of compute and API costs required to replicate results?**
>
> This is a valuable question for reproducibility. Each full evaluation run (one model, one scaffold, public set) requires roughly 731 agent trajectories with up to 250 turns each, using frontier API access. We will include an appendix section in the final paper with an estimated cost breakdown per model and scaffold, to help the community plan replication efforts.
>
> **What are the institutional concerns that might arise around maintaining private/heldout sets going forward? Are these major concerns for these kinds of work?**
>
> The held-out set is maintained by the benchmark authors and is not released publicly, aimed at mitigating the risk of community-level leakage. The commercial set is governed by binding agreements with startup partners, limiting the release of the private codebases. The primary institutional risk is organizational continuity (ensuring the benchmark remains actively maintained). Since the release of the benchmark, we have actively maintained both the public and the private sets including updating results for latest models, and plan to continue to do so in the future. We acknowledge this as a genuine limitation and will discuss it explicitly in the limitations section.

---

> > ### Author Rebuttal · Reviewer_cYcP · 2026-04-01
> >
> > Thanks to the authors for these answers. I think including these in appendices will further amplify impact of the work. I remain very positive about this submission.

---

### Decision · Program_Chairs · 2026-04-30

**Decision:**

Accept (regular)

**Comment:**

The reviewers agreed that the paper proposes an interesting benchmark for software engineering tasks that capture more realistic, long-horizon tasks. The extensive empirical evaluation provides insights into the current capabilities of AI agents for complex software engineering tasks, and also demonstrates the usefulness of this benchmark for future research. However, the reviewers also raised several concerns and questions in their initial reviews, particularly regarding institutional concerns in maintaining the benchmark, potential biases introduced by human augmentation of tasks, claims of contamination resistance, and potential distribution mismatches across different task sets.

We want to thank the authors for their detailed responses and active engagement with the reviewers. The reviewers appreciated the responses, which helped in answering their key questions. Based on the authors' responses and follow-up discussions, the reviewers are generally positive, and the recommended decision is to accept the paper. The reviewers have provided detailed feedback, and we strongly encourage the authors to incorporate this feedback when preparing an updated version of the paper.

However, we also noticed that some references in the paper have issues, including:

- The reference "A survey on data contamination for large language models. arXiv preprint arXiv:2502.14425, 2025" has a mismatch in the names of some authors.

- The reference "Benchmark data contamination of large language models: A survey. arXiv preprint arXiv:2406.04244, 2024" has a wrong list of authors.

- The reference "Multi-swebench: A multilingual benchmark for issue resolving. arXiv preprint arXiv:2404.02605, 2024." has an error in the arXiv ID and year.

It is important that the authors carefully check all the references in the paper and correct any such issues.